# Impact of Heavy Metals on Glioma Tumorigenesis

**DOI:** 10.3390/ijms242015432

**Published:** 2023-10-21

**Authors:** Gerardo Caruso, Aristide Nanni, Antonello Curcio, Giuseppe Lombardi, Teresa Somma, Letteria Minutoli, Maria Caffo

**Affiliations:** 1Department of Biomedical and Dental Sciences and Morphofunctional Imaging, Unit of Neurosurgery, Università degli Studi di Messina, 98125 Messina, Italy; nnnrtd93r21l112i@studenti.unime.it (A.N.); crcnnl91s28e977t@studenti.unime.it (A.C.); maria.caffo@unime.it (M.C.); 2Department of Oncology, Oncology 1, Veneto Institute of Oncology IOV-IRCCS, 35128 Padua, Italy; giuseppe.lombardi@iov.veneto.it; 3Division of Neurosurgery, Department of Neurological Sciences, Università degli Studi di Napoli Federico II, 80125 Naples, Italy; teresa.somma@unina.it; 4Department of Clinical and Experimental Medicine, Università degli Studi di Messina, 98125 Messina, Italy; letteria.minutoli@unime.it

**Keywords:** heavy metals, free radicals, glioma, oxidative stress, reactive oxygen species

## Abstract

Recently, an increase in the incidence of brain tumors has been observed in the most industrialized countries. This event triggered considerable interest in the study of heavy metals and their presence in the environment (air, water, soil, and food). It is probable that their accumulation in the body could lead to a high risk of the onset of numerous pathologies, including brain tumors, in humans. Heavy metals are capable of generating reactive oxygen, which plays a key role in various pathological mechanisms. Alteration of the homeostasis of heavy metals could cause the overproduction of reactive oxygen species and induce DNA damage, lipid peroxidation, and the alteration of proteins. A large number of studies have shown that iron, cadmium, lead, nickel, chromium, and mercury levels were significantly elevated in patients affected by gliomas. In this study, we try to highlight a possible correlation between the most frequently encountered heavy metals, their presence in the environment, their sources, and glioma tumorigenesis. We also report on the review of the relevant literature.

## 1. Introduction

The environmental factors potentially linked to the increase in the risk of the onset of neoplastic pathologies are evidently connected to the degree of industrialization. High urbanization and an ever-increasing number of individuals employed in particular work activities have certainly increased the risk of contact and accumulation of heavy metals in the body and, consequently, the risk of the onset of cerebral gliomas (Table 1) [1]. Given the development of various industries (metal plating facilities, mining operations, etc.), metals in wastewater continue to be directly or indirectly discharged into the environment. 

Substances that promote cancer (carcinogens) can act directly or indirectly on DNA, causing mutations [2]. The mechanism of action of these substances can be genotoxic or non-genotoxic. Genotoxic carcinogens bind directly to DNA, causing mutations. They are dangerous even at low doses, as the carcinogenicity does not have a margin of tolerance. Non-genotoxic carcinogens affect processes regulated by or dependent on DNA and gene expression. For these molecules, their use is tolerated unless the exposure or intake level exceeds the threshold values [3]. Although the molecular mechanisms are not completely clear, the potential of heavy metals to generate reactive oxygen species (ROS) and alter cellular redox states is considered the key mechanism in the induction of carcinogenesis. Carcinogens can thus enter the human body through respiration, entering the bloodstream at the level of the alveoli, through the blood reaching and accumulating in various tissues, including the central nervous system (CNS). Some heavy metals can cross the blood–brain barrier (BBB), accumulating in the brain parenchyma, thus causing cellular damage and potentially triggering malignant transformation [4].

Gliomas are the most frequent primary malignant brain tumors in the adult population [5]. According to the most recent classification of the World Health Organization (WHO), diffuse gliomas in adults are classified into the following: glioblastoma (GB) IDH-wildtype, astrocytoma IDH-mutant, and oligodendroglioma IDH-mutant, as well as 1p/19q codeleted [6]. Glioblastoma IDH-wildtype is the most common malignant primary tumor in the CNS, accounting for 14.5% of all CNS tumors and 48% of malignant CNS tumors [7]. Glioblastoma shows a dismal prognosis, with a reported median overall survival (OS) of approximately 15 months [8]. Surgical treatment followed by radiotherapy and chemotherapy represents the standard of care. Improvements in surgical techniques, including the intraoperative mapping of eloquent areas of the brain and the use of fluorescent dyes that are helpful in the detection of tumor borders, show only a slight increase in survival [9]. Nonetheless, despite the continuous technological progress related to surgical procedures and the advent of new therapeutic protocols, the prognosis of patients with gliomas remains decidedly poor. Moreover, the etiopathogenetic factors linked to the onset of cerebral gliomas are still poorly understood.

In this review, we want to focus on the possible interactions that could occur between heavy metals and brain gliomas.

## 2. Gliomas

Surgical treatment followed by radiotherapy and chemotherapy represents the most common treatment for patients harboring gliomas. Improvements in surgical techniques, including intraoperative mapping of eloquent areas of the brain and the use of fluorescent dyes that are helpful in the detection of tumor borders, show only a slight increase in survival [10]. Gliomagenesis is characterized by numerous biological events, such as activated growth factor receptor signaling pathways, the downregulation of many apoptotic mechanisms, and an imbalance among proangiogenic and antiangiogenic factors [11]. Several growth factor receptors, including epidermal growth factor receptor (EGFR), platelet-derived growth factor receptor (PDRGF), and vascular endothelial growth factor receptor (VEGFR), are dysregulated in gliomas. 

### Genetic Alterations in Gliomas 

Epigenetic abnormalities frequently affect many biological mechanisms, including cellular cycle regulation. The dysregulation of TP53, retinoblastoma (Rb), inhibitor of cyclin-dependent kinase 4a (INK4a), and phosphatase and tensin homolog (PTEN) genes is typical of glioma cells. These alterations can cause cell cycle alterations (growth, differentiation, and function). Epigenetic alterations in neoplastic lesions are DNA methylation, alterations of nucleosomes and histones, and acetylation processes. DNA methylation involves the methyl group at the C-5 position of cytosine and commonly occurs in the CpG nucleotide [12]. DNA hypermethylation triggers silencing genes capable of interfering with various biological processes such as apoptosis, angiogenesis, and DNA repair mechanisms [12]. Primary GBs show demethylation, dysregulation of the MAGEA1 oncogene and EGFR, loss and/or mutation of homozygous cyclin-dependent kinase (CDK) inhibitor p16INK4A/(CDKN2A), alterations in the tumor suppressor PTEN on chromosome 10, and deletion in the INK4a gene with losses of p14 and p16 [13]. The progression from low-grade to malignant gliomas involves mutations of the tumor-suppressor gene TP53 and higher expressions of PDGF ligands and receptors; it also involves the accumulation of genetic alterations in retinoblastoma-associated cell-cycle regulatory pathways, including the dysregulation of p16INK4A/CDKN2A or the retinoblastoma susceptibility locus 1 (pRB1), as well as over-expression of cyclin-dependent kinase 4 (CDK4) and human double minute 2 (HDM2). The evolution to secondary GB is associated with the deletion of chromosome 10, which includes tumor-suppressor phosphatase and tensin homolog (PTEN). The astrocyte elevated gene-1 (AEG-1) and Ha-ras family of retrovirus-associated DNA sequences (RAS) are amplified in gliomas, inducing neoplastic progression [14]. Furthermore, oncogenic Ha-ras induces AEG-1 expression by modulating the phosphatidylinositol 3-kinase (PI3K)-Akt signaling pathway and contributes to the proliferation of gliomas. Mitogenetic signals activate a molecular cascade known as ras-mitogen-activated protein kinase (Ras/MAPK). MAPK inhibits the Rb gene and activates the transcriptional factor E2F, and cells enter the S phase. The INK4a gene influences the Rb pathway by activating three cyclin kinase inhibitors: p15, p16, and p19. The final result is the blockage of cyclindependent kinases 2, 4, and 6, triggering cell cycle progression by inhibiting pRb [15]. The MGMT gene regulates a DNA repair enzyme that removes alkyl adducts from the O6-position of guanine. Methylation of the MGMT gene’s promoter makes cancer cells more responsive to the alkylating agent’s effects and, at the same time, represents a predictive factor of favorable survival in GB patients [16]. The IDH1 mutation causes the inhibition of DNA demethylation and the accumulation of methylated DNA [17]. 

Epigenetic alterations of histones affect the integrity of the genome and the genic expression. Histones are nuclear proteins that package DNA into nucleosomes. The Nterminal tracts of histones are subject to many modifications, such as acetylation, methylation, phosphorylation, and ADPribosylation. Frequently, in the genomic analysis of GB, in response to alteration of the regulatory genes, aberrations of the histone deacetylases 2 and 9 (HDAC2 and HDAC9) occur. In GBs, histone 3 appears significantly acetylated, and the epigenetic alteration of the BMI-1 protein, which regulates histone H3K27, is associated with a worse prognosis [18,19]. The H3F3A/G34 alteration, present in some biomolecular variants of GBs, is linked to a high percentage of mutation in P53, ATRX, and DAXX [20].

MicroRNAs are nucleotide sequences capable of modifying gene expression by interacting with mRNA. In various types of neoplasms, their alteration has been demonstrated, causing, in these cases, epigenetic anomalies [21,22]. The increased expression of MIR-21, blocking the apoptotic mechanisms, seems to be responsible for the progression of GB [23]. Furthermore, it has also been demonstrated that the hyperactivity of MIR21 is able to inhibit metalloproteases and alter the regulation of genes responsible for controlling the migration and apoptosis of tumor cells [24].

## 3. Heavy Metals

Reduced doses of the various chemical elements are essential for the numerous processes that take place within the cells of the organism. On the contrary, high concentrations of the same elements can be responsible for important toxic effects and, through the activation of ROS, have carcinogenic activity [25]. The objective of this review is to hypothesize the possible connection between exposure to heavy metals and the risk of the onset of brain tumors. The main limitation is the reduced availability of healthy brain tissue compared with tumor tissue [26]. However, numerous pieces of scientific evidence have demonstrated that prolonged exposure to heavy metals can cause toxic effects on humans [27]. These elements can cause alterations in the mechanisms related to cellular metabolism. Furthermore, excessive production of ROS causes DNA damage and induces the formation of cancers [28]. The activation of ROS in the pathogenesis of cerebral gliomas is now widely studied. Super-oxide dismutase and catalase (CAT) activity are particularly studied in brain tumors [29].

### 3.1. Heavy Metals and Oxidative Stress

Heavy metals are defined as those metals having a density higher than 5.0 g/cm^3^, a low solubility of their hydrates, a typical behavior like cations, a strong ability to form complexes, and an affinity toward sulphides. They are all “transition metals” (chemical elements that form ions only partially filled with electrons because their internal orbits are not complete with electrons). This group includes chemical elements such as iron (Fe), chromium (Cr), mercury (Hg), nickel (Ni), lead (Pb), copper (Cu), aluminum (Al), cadmium (Cd), and some metalloids with properties similar to those of heavy metals, such as arsenic (As), bismuth (Bi), and selenium (Se). It is already known that metals induce toxicity and carcinogenicity, contributing to the formation of reactive oxygen and nitrogen species and free radicals, which include numerous compounds (ROS, reactive for oxygen and nitrogen, superoxides, containing superoxide anion O_2_-, hydroxyl OH-, nitric oxide NO-, nitrogen dioxide NO_2_-, peroxide, and singlet oxygen O_2_+), all aggressive substances, and powerful oxidants. Although the exact mechanisms are still unclear, new data suggest that heavy metals interfere with the redox state, representing a potential mechanism related to carcinogenesis [30]. As a result of the oxidative state, there is an excessive endogenous production of free radicals. Oxidative stress can cause DNA mutations, genomic instability, strand breaking, and cell death (Figure 1). The formation of free radicals is the cause of alterations in proteins structures, altering the calcium circulation and homeostasis of sulfur bridges. The oxidative action on phospholipids (polyunsaturated fatty acids present in cell membranes) and on the easily oxidizable LDL fraction of circulating cholesterol leads to the production of oxidized lipid molecules (free radicals). The action of these elements on proteins, particularly those of the vascular endothelium, induces the process of atherosclerosis, while the metabolism of calcium, essential for the transmission of intracellular stimuli, is altered with the consequent reduction of the cellular secretory capacity. Finally, metals, which have a particular affinity for sulfur molecules, bind to the sulfhydryl radicals present in the amino acids that form proteins, blocking the functions for which these structures are intended. Proteins that contain sulfur have important functions, including the removal of toxic substances and antioxidant and enzymatic functions.

ROS have the function of intracellular messengers, and, consequently, reduced levels can be responsible for altered cellular biological processes. In the second step of metal carcinogenesis, through phosphorylation, ROS alter the structure of signal proteins. Furthermore, through apoptotic or oxidative mechanisms, they can interfere with the normal transduction pathways. DNA appears to be highly susceptible to ROS-induced stress. As a consequence of this event, single- or double-strand breakage, base modifications, deoxyribose modification, and DNA cross-linking are frequently encountered. The metal-transformed cells have an autophagy deficiency resulting in the accumulation of p62 and constitutive activation of NF-E2-related factor 2 (Nrf2), which leads to higher levels of antioxidants, increased levels of B cell lymphoma 2 (Bcl-2), inflammation, and angiogenesis [31,32]. A possible explanation of this complex process is that, in a first step, ROS induce the neoplastic transformation in the normal cell. In the following phase, the neoplastic cell can exploit the anti-oncogenic characteristics of ROS, creating an appropriate microenvironment with reduced levels of ROS, such as to allow the same neoplastic cells to escape the apoptotic mechanisms [33]. ROS can also trigger or favor neoplastic proliferation if the mutations affect oncogenes or tumor suppressor genes [34].

### 3.2. Heavy Metals and Gliomas

The considerable need for oxygen (20%), the presence of polyunsaturated fatty acids in cell membranes, the high percentage of iron, and the reduced activity of antioxidant enzymes make the brain highly susceptible to oxidative stress. Protein oxidation and lipid peroxidation have been detected in the hippocampus and neocortex of patients with Alzheimer’s disease, as well as within the motor neurons in amyotrophic lateral sclerosis. Furthermore, ROS, through apoptotic mechanisms, can induce the death of neurons and astrocytes. Recently, it has been proven that TPA-induced invasion/migration of U87 cells were inhibited by protein kinase C (PKC) inhibitors, Go6976, COX-2 inhibitors, NS398, NADPH oxidase inhibitors, diphenyleneiodonium chloride (DPI), and the ROS scavengers superoxide dismutase (SOD) and tempol [35]. Thus, the stimulation of cyclooxygenase-2/prostaglandin E2 and metalloproteinase-9 via ROS-activated ERKs is involved in the invasion/migration of U87 glioma cells elicited via TPA and antioxidative substances such as quercetin, baicalein, and myricetin, causing the suppression of invasion/migration processes in glioma cells. Microglia are parenchymal cells capable of antigen presentation to T-cells that patrol the CNS. Microglia seem to facilitate glioma invasion by digesting extracellular matrix components and restricting tumor cell motility [36]. Activated microglia release oxygen metabolites, reactive nitrogen species, and proteinases. Autopsy studies have demonstrated, in adults exposed to heavy metals, an increase in CD14 as an expression of monocyte infiltration. Microglial activation with manganese chloride induces neurotoxicity in vitro, and the use of antioxidants such as superoxide dismutase/catalase, glutathione, or inhibitors of NO biosynthesis effectively protects dopaminergic neurons. 

Steenland and Boffetta demonstrated an increased risk of the onset of brain tumors in individuals subjected to prolonged exposure to metals for occupational reasons [37]. In a research study, the potential carcinogenetic role of metals such as nickel, cadmium, chromium, arsenic, silicon, and beryllium in human brain tumors was investigated. A statistically significant association was inferred between the development of cerebral tumors and the concentrations of silicon (*p* = 0.01), magnesium (*p* = 0.01), calcium (*p* = 0.03), and zinc (*p* = 0.05). In a recent study, analysis of serum concentrations of metals, including Zn, Pb, Co, Cd, and Fe, was performed via spectrophotometric examination. Serum Cd, Fe, Mg, Mn, Pb, and Zn levels were increased in patients compared to the control group [38]. Recently, Gaman et al. demonstrated the presence, in primary brain tumor tissue, of chemical elements including Cu, Fe, Cd, Al, Hg, Ni, and Pb [39]. Nguyen highlighted that heavy metals, especially arsenic, mercury, lead, and cadmium, are able to modify the etiology of gliomas. The author showed that five genes (SOD1, CAT, GSTP1, PTGS2, TNF), two miRNAs (hsa-miR-26b-5p and hsa-miR-143-3p), and transcription factors (DR1 and HNF4) are possible key components related to combined heavy metals and glioma development [40]. On the other hand, Xie et al. did not find a direct link between concentrations of heavy metals and biomarkers of gliomas such as tumor grade, P53, and Ki-67 [41]. Glioma patients had high levels of serum ferritin. Glial cells, due to the increase in ferritin and transferrin receptor 1 values, increase the intracellular absorption of iron [42]. This mechanism favors the phenomenon of ferroptosis, an iron-dependent form of nonapoptotic cell death. It is characterized by the accumulation of lipid peroxidation products and ROS derived from increased iron metabolism. Within the cytoplasm, chromium is reduced to chromium (III), resulting in the production of ROS. Two studies evaluated the impact of occupational exposure and the risk of the onset of brain tumors without, however, finding any statistical significance [43,44]. An epidemiologic study has instead demonstrated that chromium exposure represents a risk factor for lung cancer, malignant lymphoma, and brain tumors [45]. Lead could promote carcinogenesis by altering DNA repair processes and interfering with the synthesis of tumor suppressor proteins [46]. Recent evidence suggests that lead is able to cross the barrier, causing an increase in its values in the brain parenchyma. In another epidemiological study, the risk of developing brain tumors as a result of continued exposure to lead was evaluated using standardized mortality ratios, proportional hazards, and Poisson regression techniques, adjusting for the effects of age, gender, and other covariates. The data obtained show an increased likelihood of brain tumor incidence in those most exposed to lead. In addition, case–control studies of occupational exposures to lead report a slight increased risk of cerebral tumors at the highest levels of lead [47]. Cadmium is a heavy metal commonly found in the earth’s crust, combined with other elements such as oxygen, chlorine, or sulfur. However, having been released into the environment for decades via anthropogenic activities, as a non-genotoxic agent, it cannot directly cause DNA mutations; instead, it can act through the alteration of epigenetic mechanisms and gene expression, the induction of oxidative stress, the inhibition of DNA repair mechanisms, and the interaction with proteins involved in cell cycle control, apoptosis, and cellular defense systems. ROS may be involved in cadmium-induced genotoxicity and carcinogenicity. Furthermore, since mitochondria are the primary sites of ROS production, these organelles and their functions seem to be implied in these processes. A recent study confirms the carcinogenic effects of cadmium, showing its effects on increasing the permeability of the BBB. The mechanisms involved in arsenic carcinogenesis are not clear. Arsenic forms of ROS under physiological conditions directly bind with critical thiols. As a carcinogen, it acts through epigenetic processes. The carcinogenic potential of arsenic may be attributed to activation of redox-sensitive transcription factors and other pathways involving nuclear factor κB, activator protein-1, and p53 [48]. Various observations have demonstrated that prolonged contacts to arsenic or its compounds are correlated to major risks of lung, skin, liver, bladder, and brain cancers. Mercury is an environmental toxicant that is correlated with brain toxicity. Humans can be exposed to methylmercury (MeHg), a neurotoxic organic form of mercury, by consuming contaminated seafood. MeHg shows genotoxic activity, causing damage to the CNS. However, the potential carcinogenic activity of mercury in brain tumors is not well demonstrated. Copper is instrumental in the activity of a number of metalloenzymes, and a number of tissue abnormalities and disease states in humans have been associated with either reduced or elevated levels of copper. Copper stimulates cancer cell proliferation, regulates oxidative phosphorylation, and increases the dependence on glycolysis in copper-deficient cancer cells in animal models [49]. The reduction to copper2+ induces the formation of ROS, which could increase in cases of increased copper absorption. However, the concentration of copper varies according to the neoplastic tissue; it is the lowest in meningeal tumors and the highest in malignant neuroectodermal tumors. Nickel exposure causes oxidative stress and the formation of free radicals in various tissues, which cause alterations of various DNA bases, enhanced lipid peroxidation, altered calcium and sulphydryl homeostasis, and increased expression of p53, NF-kβ, AP-1, and MAPK. Prolonged exposure to nickel is strongly correlated with an increased likelihood of malignancy and, specifically, brain tumors [50]. Aluminum can bind to negatively charged brain phospholipids, which contain polyunsaturated fatty acids and are easily attacked via reactive oxygen species (ROS) [51]. Aluminum can also stimulate iron-initiated lipid peroxidation in the Fenton reaction, which causes ROS production and Fe^3+^ formation.

#### Metal Nanoparticles

Nanoparticles (NPs) are particles with at least one dimension <1 μm and potentially as small as atomic- and molecular-length scales (~0.2 nm) [52]. Although NP applications have great potential, there are concerns about their adverse effects on human health and the environment. The same properties (size, shape, and chemical composition) of the particle can have an impact on the ecosystem [53]. Exposure to air pollution has numerous health effects, due to the growing trend of pollutants. Environmental NPs are byproducts of combustion processes (combustion of fossil fuels, petroleum, and wood burning), automobile exhaust, the activity of volcanoes, or industrial derivatives [54]. Industrial NPs are frequently composed of heavy metals and their metal oxides [55]. Metal NPs can enter the body mainly through the respiratory, dermal, gastrointestinal, circulatory, immunological, and neurological tracts. The action of NPs is initially due to direct contact with cells (primary damage), causing toxicity, oxidative stress, DNA damage, and inflammation. The NPs themselves, due to their peculiar properties, enter the blood circulation and, consequently, can also accumulate in the various organs, causing inflammation while altering, at the same time, their physiological functions (secondary damage). Environmental pollutants may reach the CNS through the transmission of NPs directly from the sinuses to the brain tissue through the olfactory nerves, the passage of NPs through the alveoli, and the blood–brain barrier [56]. Toxicity from metal NPs depends on its physical and chemical properties: molecular shape, size, oxidation status, surface area, bonded surface species, surface coating, solubility, and the degree of aggregation cause the generation of ROS and toxicity [57]. The induction of oxidative stress, as a consequence of the generation of ROS, activates a series of inflammatory mediators, such as nuclear factor-κB (NF-κb), mitogen-activated protein kinase (MAPK), and the phosphoinositide 3-kinase (PI3K) pathway [58]. It has been reported that nano-CuO shows notable cytotoxic and DNA-damaging capacity, leading to 8-hydroxy-20-deoxyguanosine (8-OHdG) formation [59]. Metal oxide NPs, including ZnO, Fe_3_O_4_, Al_2_O_3_, and CrO_3_, with particle sizes ranging from 30 to 45 nm, have been reported to induce apoptosis [60]. Cadmium telluride quantum dots were found to significantly increase p53 levels and upregulate the p53-downstream effectors Bax, Puma, and Noxa [61]. Metal oxide NPs such as cadmium and iron have also been reported to show toxic effects via the induction of inflammatory-related cytokine release induced by NF-κB.

## 4. Heavy Metals and Other Brain Tumors

Meningiomas are the most frequent primary tumors of the central nervous system [62], representing approximately 35% of all primary CNS tumors [63]. Ionizing radiation appears to be, to date, the only environmental risk factor [64]. Already in 1981, Muller and Iffland demonstrated the presence, through the use of atomic absorption spectrometry, of Cr, Fe, Cu, Ca, and other metals in a series of 30 meningiomas [65]. Nagaishi et al. reported two cases of angiomatous meningioma, characterized by the presence of iron deposits in the cytoplasm of tumor cells. The authors noted that the presence of iron deposits is typical of highly vascularized meningiomas and is correlated with increased oxidative DNA damage markers [66]. Various epidemiological studies have found a possible correlation between work activities that involve the use of metallic elements and meningiomas [67,68]. Hu et al. demonstrated the association between the risk of meningiomas and occupations related to the use of lead, cadmium, tin, and ionizing radiation [69]. Similarly, Liao et al. demonstrated the possible association between lead exposure and the risk of meningiomas [70]. The analysis of the data from the INTEROCC study demonstrated the positivity of exposure to iron and chromium and the risk of meningioma [71]. Iron auto-oxidation, activation of oxidative-responsive transcription factors and proinflammatory cytokines, and iron-induced hypoxia signaling are possible mechanisms capable of triggering the tumor [72]. In support of this, Rajaraman et al. demonstrate the association between a functional polymorphism in the heme synthesis pathway and meningioma risk. The authors have, in fact, demonstrated that the ALAD2 allele of the G177C polymorphism was associated with an increased risk of meningioma but not of glioma [73]. A recent meta-analysis has highlighted how lead may represent a risk factor for meningiomas and brain tumors. The same authors have also demonstrated a higher incidence of brain tumors in children of parents exposed to lead [74]. Pituitary adenomas are tumors, generally benign, of the pituitary gland. Recently, it has been hypothesized that heavy metals can also increase the risk of the onset of pituitary adenomas [75]. It is probable that, on the basis of this, there is a possible interaction between the AIP (Aryl hydrocarbon receptor-interacting protein)-AhR pathway and heavy metals. AIP mutations are found more frequently in adenomas, which are more invasive and resistant to medical treatments [76]. Exposure of Noble rats to high cadmium exposure is correlated with a significant increase in pituitary adenomas [77]. It is probable that the action of cadmium is expressed through the aforementioned AIP-AhR pathway [78]. Brain metastases have an incidence of 30%, appear to be progressively and constantly increasing, and are the most frequent brain neoplastic lesions in the adult population [79]. Heavy metals are able to inhibit the expression of metalloproteinases (TIMPs), a tumor suppressor gene, thus favoring the onset of brain metastases in cell lines [80]. Haga et al. reported that cadmium, when present in biological systems, is often linked to metabolic proteins, such as metallothioneins or heat shock proteins, closely related to metastasis [81]. Some heavy metals (cadmium, copper) are able to stimulate angiogenesis processes, such as up-regulation of vascular endothelial growth factor secretion and endothelial cell proliferation and migration processes [82]. Some authors have found nickel accumulations whose extent depends on the tumor histotype. In this way, the different concentrations of nickel in primary brain tumors and metastases appear to be more understandable [83,84]. Recently, Gaman et al. have demonstrated the presence of heavy metals, in different concentrations, in the tissues of brain tumors and brain metastases [39].

## 5. Conclusions

Gliomagenesis is a complex process characterized by the triggering of numerous pathways and many molecules. The genetic alterations that seem to induce the development of the neoplasm are various, and not all have yet been demonstrated and clearly elucidated. Gliomagenesis itself still appears to be incompletely understood. The new pharmacological therapies, precisely by virtue of the high number of interesting molecules, are not able to effectively counteract tumor progression. In fact, the generally accepted therapeutic protocol provides for surgical resection of the tumor, followed by radiotherapy and/or chemotherapy. Despite the advent of new technologies, the result obtained is only a modest increase in survival, while the prognosis remains substantially poor.

Oxidative stress appears to be linked to carcinogenesis, including gliomagenesis. ROS can interact with DNA by directly altering it or, indirectly, by interfering with the repair mechanisms. Although the exact molecular mechanisms are unclear, new research suggests that heavy metals can interfere with the redox state, representing a potential mechanism related to gliomagenesis. Excessive endogenous production of free radicals can cause DNA mutations, genomic instability, strand breaking, and cell death.

In this review, the possible connections between heavy metals and the risk of the onset of cerebral gliomas are reported through a rereading of the relevant literature. Humans are exposed to numerous chemical elements, and scientific results suggest that some of these compounds are neurotoxicants, causing the onset of brain tumors. This is due to the lack of sufficient information and regulation on the thousands of substances that are continuously released into the environment. Furthermore, these findings are limited by the small sample size of the groups and should be considered as hypothesis generators for future research. Further studies are crucial to understanding the potential brain cancer risk associated with environmental carcinogens as well as their biological mechanisms of action. 

## Figures and Tables

**Figure 1 ijms-24-15432-f001:**
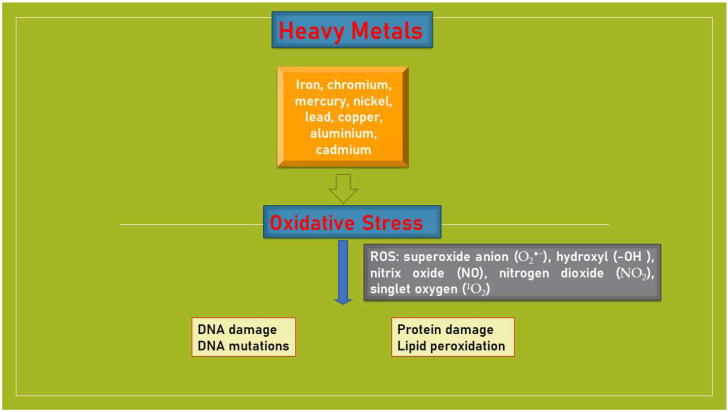
Schematic representation of heavy metals in relation to production of ROS.

**Table 1 ijms-24-15432-t001:** Heavy metals and sources of exposure.

Heavy Metals	Atomic Number	Sources of Exposure
Iron (Fe)	26	Household appliances, utensils, food containers, motor vehicles, hulls, armaments, and in steel and cast-iron alloys
Chromium (Cr)	24	Enamels, paints, dyes, leather tanning, and fabric dyes
Mercury (Hg)	80	Detonators, pigments for antifouling paints (ship hulls), pesticides, for the preparation of soda in electrolytic cells, in the manufacture of physics apparatus (barometers and pressure gauges), batteries lamps, to make mirrors, and in dentistry
Nickel (Ni)	28	Rechargeable batteries, ship propellers, kitchen equipment, industrial chemical plant pipes, coins, coatings of iron, brass, and other metallic materials
Lead (Pb)	82	Construction, in the production of batteries, enamels for pottery, crystal glass, in the automotive sector, in bullets for firearms, and in the liquid state, as a coolant in some types of nuclear reactors
Copper (Cu)	29	Electric cables, pipes, furnishing finishes, sculptures, pictorial support, coins, musical instruments, dyes, pots, chemical solutions, fungicides, and dissipators
Aluminum (Al)	13	Transportation, packaging, construction, consumer durables, power lines, optics, and firearms or parts thereof—ammunition shells and shells
Cadmium (Cd)	46	Electrodes, coatings of metallic materials, solders, and pigments
Arsenic (As)	33	Insecticides, cosmetics, fireworks, photovoltaic panels, and dyes

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
