# Peer review of "Impact of Heavy Metals on Glioma Tumorigenesis"

_ijms, 2023, doi:10.3390/ijms242015432_

Round 1

Reviewer 1 Report

This review article elaborates on the connection between heavy metal exposure and gliomagenesis. The authors cite past studies that have shown that the levels of several heavy metals encountered in the environment can be significantly elevated in patients with malignant gliomas compared to controls.

The authors also discuss the role of the blood-brain barrier, ROS species, and the major genetic and epigenetic changes that occur in gliomagenesis. The review is informative and highlights the importance of further research to understand the biological role of heavy metals in glioma.

Minor suggestions:

- The introductory sections on heavy metals and gliomas can be shortened, adding emphasis to the role of heavy metals in this cancer type.

- Previous studies have also reported a connection between heavy metal exposure and other types of brain tumors. This needs to be discussed along with similarities and dissimilarities between how different brain tumor types are affected by heavy metals.

- An illustrative image should make the review easier to read.

Author Response

We thank Reviewer for suggestions. We have carefully read reviewer’s comments, and have revised our manuscript accordingly.

The introductory sections on heavy metals and gliomas can be shortened, adding emphasis to the role of heavy metals in this cancer type.

  • In accordance with your suggestion, we have modified the introductory section in the revised manuscript.

The increase in cancer burden due to environmental factors is indeed connected to the degree of industrialization. A greater urbanization rate, a greater number of adult populations at risk due to exposure to certain lifestyles, and a greater number of workers assiduously exposed to environmental carcinogens constitute significant cancer risk factors (Table 1) [1]. Given the development of industries (metal plating facilities, mining operations, etc.), metals in wastewater continue to be directly or indirectly discharged into the environment. Substances that promote cancer (carcinogens) can act directly or indirectly on DNA, causing mutations [2]. The mechanism of action of these substances can be genotoxic or non-genotoxic. Genotoxic carcinogens bind directly to DNA, causing mutations in the genetic material. They are dangerous even at low doses, as the carcinogenicity does not have a margin of tolerance.  In contrast, non-genotoxic carcinogens affect processes regulated by or dependent on DNA and gene expression. For these molecules, there is a margin of safety; thus, their use is tolerated unless the exposure or intake level exceeds the threshold values [3]. Although the molecular mechanisms are not completely clear, the potential of heavy metals to generate reactive oxygen species (ROS) and alter cellular redox states is considered the key mechanism in metal-induced carcinogenesis. Carcinogens can thus enter the human body through respiration, entering the bloodstream at the level of the alveoli, and through the blood reaching and accumulating in various tissues, including the central nervous system (CNS). It is important to highlight that the CNS is protected by the blood-brain barrier (BBB), which makes the brain an immunologically privileged environment. Nonetheless, some heavy metals can cross the BBB, accumulating in the brain parenchyma for long periods, thus causing cellular damage and potentially triggering malignant transformation [4]. Gliomas are the most frequent primary malignant brain tumors in the adult population [5]. According to the most recent classification of the World Health Organization (WHO), diffuse gliomas in the adult are classified into: glioblastoma (GB) IDH-wildtype, astrocytoma IDH-mutant and oligodendroglioma IDH-mutant and 1p/19q codeleted [6]. Glioblastoma IDH-wildtype is the most common malignant primary tumor in the CNS, accounting for 14.5% of all CNS tumors and 48% of malignant CNS tumors [7]. Glioblastoma shows a dismal prognosis, with a reported median overall survival (OS) of approximately 15 months [8]. Surgical treatment followed by radiotherapy and chemotherapy represent the standard of care. Improvement in surgical techniques, including intra-operative mapping of eloquent areas of the brain and the use of fluorescent dyes that are helpful in the detection of tumor borders, show only a slight increase in survival [9]. Nonetheless, the continuous technological progress related to surgical procedures and the advent of new therapeutic protocols, the prognosis of patients with gliomas remains decidedly poor. Moreover, the etiopathogenetic factors linked to the onset of cerebral gliomas are still poorly understood. In this review, we want to focus on the possible interactions that could occur between heavy metals and brain gliomas.

  1. 1. Schottenfeld, D.; Beebe-Dimmer, J.L.; Buffler, P.A.; Omenn, G.S. Current Perspective on the Global and United States Cancer Burden Attributable to Lifestyle and Environmental Risk Factors. Annu. Rev. Public Health 2013, 34, 97-117. doi: 10.1146/annurev-publhealth-031912-114350.
  2. Williams, G.M. Mechanisms of Chemical Carcinogenesis and Application to Human Cancer Risk Assessment. Toxicology 2001, 166, 3-10. doi: 10.1016/s0300-483x(01)00442-5.
  3. Nohmi, T. Thresholds of Genotoxic and Non-Genotoxic Carcinogens. Toxicol. Res. 2018, 34, 281-290. doi: 10.5487/TR.2018.34.4.281.
  4. Gilani, S.R.; Zaidi, S.R.; Batool, M.; Bhatti, A.A.; Durrani, A.I.; Mahmood, Z. Report: Central Nervous System (CNS) Toxicity Caused by Metal Poisoning: Brain as a Target Organ. Pak. J. Pharm. Sci. 2015, 28, 1417–1423.
  5. Ostrom Q.T., Gittleman H., Liao P., Vecchione-Koval T., Wolinsky Y., Kruchko C., Barnholtz-Sloan J.S. CBTRUS Statistical Report: Primary Brain and Other Central Nervous System Tumors Diagnosed in the United States in 2010–2014. Neuro-Oncology 2017, 19, v1–v88. doi: 10.1093/neuonc/nox158.
  6. Louis, D.N.; Perry, A.; Wesseling, P.; Brat, D.J.; Cree, I.A.; Figarella-Branger, D.; et al. The 2021 WHO Classification of Tumors of the Central Nervous System: A Summary. Neuro Oncol. 2021, 23, 1231-1251. doi: 10.1093/neuonc/noab106.
  7. Ostrom, Q.T.; Gittleman, H.; Truitt, G.; Boscia, A.; Kruchko, C.; Barnholtz-Sloan, J.S. CBTRUS Statistical Report: Primary Brain and Other Central Nervous System Tumors Diagnosed in the United States in 2011-2015. Neuro Oncol. 2018, 20, iv1-iv86. doi: 10.1093/neuonc/noy131.
  8. Grochans, S.; Cybulska, A.M.; Simińska, D.; Korbecki, J.; Kojder, K.; Chlubek, D.; Baranowska-Bosiacka, I. Epidemiology of Glioblastoma Multiforme-Literature Review. Cancers 2022, 14, 2412. doi: 10.3390/cancers14102412.
  9. Caruso, G.; Fazzari, E.; Cardali, S.M.; Caffo, M. Applications of Nanoparticles in the Treatment of Gliomas. In Nanoparticle Drug Delivery Systems for Cancer Treatment, 1st ed.; Gali-Muhtasib, H.; Chouaib, R.; (Eds). Jenny Stanford Publishing, New York, 2020; pp.182-216.

Previous studies have also reported a connection between heavy metal exposure and other types of brain tumors. This needs to be discussed along with similarities and dissimilarities between how different brain tumor types are affected by heavy metals.

  • In accordance with the reviewer's suggestion, we, in the revised text, have added a new section: “Heavy Metals and Other Brain Tumors"

Meningiomas are the most frequent primary tumors of the central nervous system [62], representing approximately 35% of all primary CNS tumors [63]. Ionizing radiation appears to be, to date, the only environmental risk factor [64]. Already in 1981, Muller and Iffland, demonstrated the presence, through the use of atomic absorption spectrometry, of Cr, Fe, Cu Ca and other metals in a series of 30 meningiomas [65]. Nagaishi et al., reported two cases of angiomatous meningioma, characterized by the presence of iron deposits in the cytoplasm of tumor cells. The authors noted that the presence of iron deposits is typical of highly vascularized meningiomas and is correlated to increased oxidative DNA damage markers [66]. Various epidemiological studies have found a possible correlation between work activities that involve the use of metallic elements and meningiomas [67-68]. Hu et al., demonstrated the association between the risk of meningiomas with occupations related to the use of lead, cadmium, tin and ionizing radiation [69]. Similarly, Liao et al. demonstrated the possible association between lead exposure and the risk of meningiomas [70]. The analysis of the data from the INTEROCC study demonstrated the positivity of exposure to iron and chromium and the risk of meningioma [71]. Iron auto-oxidation, activation of oxidative responsive transcription factors and proinflammatory cytokines, and iron-induced hypoxia signaling are possible mechanisms capable of triggering the tumor [72]. In support of this, Rajaraman et al. demonstrate the association between a functional polymorphism in the heme synthesis pathway and meningioma risk. The authors have, in fact, demonstrated that the ALAD2 allele of the G177C polymorphism was associated with increased risk of meningioma but not of glioma [73]. A recent meta-analysis has highlighted how lead may represent a risk factor for meningiomas and brain tumors. The same authors have also demonstrated a higher incidence of brain tumors in children of parents exposed to lead [74]. Pituitary adenomas are tumors, generally benign, of the pituitary gland. Recently, it has been hypothesized that heavy metals can also increase the risk of the onset of pituitary adenomas [75]. It is probable that at the basis of this there is the possible interaction between the AIP (Aryl hydrocarbon receptor-interacting protein)-AhR pathway and heavy metals. AIP mutations are found more frequently in adenomas, which are more invasive and resistant to medical treatments [76]. Exposure of Noble rats to high cadmium exposure is correlated with a significant increase in pituitary adenomas [77]. It is probable that the action of cadmium is expressed through the aforementioned AIP-AhR pathway [78]. Brain metastases have an incidence of 30%, appear to be progressively and constantly increasing and are the most frequent brain neoplastic lesions in the adult population [79]. Heavy metals are able to inhibit the expression of metalloproteinases (TIMPs), a tumor suppressor gene, thus favoring the onset of brain metastases in cell lines [80]. Haga et al., reported that cadmium, when present in biological systems, is often linked to metabolic proteins, such as metallothioneins or heat shock proteins, closely related to metastasis [81]. Some heavy metals (cadmium, copper) are able to stimulate angiogenesis processes, such as up-regulation of vascular endothelial growth factor secretion and endothelial cell proliferation and migration processes [82]. Some authors have found nickel accumulations whose extent depends on the tumor histotype. In this way, the different concentrations of nickel in primary brain tumors and metastases appear to be more understandable [83-84]. Recently, Gaman et al. have demonstrated the presence of heavy metals, in different concentrations, in the tissues of brain tumors and brain metastases [85].

  1. Caruso, G.; Ferrarotto, R.; Curcio, A.; Metro, L.; Pasqualetti, F.; Gaviani, P.; Barresi, V.; Angileri, F.F.; Caffo, M. Novel Advances in Treatment of Meningiomas: Prognostic and Therapeutic Implications. Cancers 2023, 15, 4521. doi.org/ 10.3390/cancers15184521.
  2. Ostrom, Q.T.; Gittleman, H.; Xu, J.; Kromer, C.; Wolinsky, Y.; Kruchko, C.; Barnholtz-Sloan, J.S. CBTRUS Statistical Report: Primary Brain and Other Central Nervous System Tumors Diagnosed in the United States in 2009–2013. Neuro Oncol. 2016, 18, v1–v75. doi: 10.1093/neuonc/now207.
  3. Greenberg, M.S. Handbook of Neurosurgery, 9th ed.; Thieme: Stuttgart, Germany, 2019; ISBN 978-1-68420-137-2.
  4. Muller, W.; Iffland, R. Studies on Metals in Meningiomas by Atomic Absorption Spectometry. Acta Neuropathol. 1981, 55, 53-58.
  5. Nagaishi, M.; Yokoo, H.; Osawa, T.; Nobusawa, S.; Tanaka, Y.; Ikota, H.; Yoshimoto, Y.; Nakazato, Y. Cytoplasmic Iron Deposition is Associated with the Expression of Oxidative DNA Damage Marker in Meningiomas. Neuropathology 2013, 33, 526–532. doi: 10.1111/neup.12023.
  6. McLaughlin, J.K., Thomas, T.L., Stone, B.J., Blot, W.J., Malker, H.S., Wiener, J.A., Ericsson, J.L. Malker, B.K. Occupational Risks for Meningiomas of the CNS in Sweden. J. Occup. Med. 1987, 29, 66–68.
  7. Preston-Martin, S., Mack, W. Henderson, B.E. Risk Factors for Gliomas and Meningiomas in Males in Los Angeles County. Cancer Res. 1989, 49, 6137–6143.
  8. Hu, J.; Little, J.; Xu, T.; Zhao, X.; Guo, L.; Jia, X.; Huang, G.; Bi, D.; Liu, R. Risk Factors for Meningioma in Adults: A Case-Control Study in Northeast China. Int. J. Cancer 1999, 83, 299–304.
  9. Liao, L.M.; Friesen, M.C.; Xiang, Y.; Cai, H.; Koh, D.; Ji, B.; Yang, G.; Li, H.; Locke, S.J.; Rothman, N.; et al. Occupational Lead Exposure and Associations with Selected Cancers: The Shanghai Men’s and Women’s Health Study Cohorts. Environ. Health Perspect. 2016, 124, 97-103. doi: 10.1289/ehp.1408171.
  10. Sadetzki, S.; Chetrit, A.; Turner, M.C.; van Tongeren, M.; Benke, G.; Fifuerola, J.; Fleming, S.; Hours, M.; Kincl, L.; Krewski, D.; et al. Occupational Exposure to Metals and Risk of Meningioma: A Multinational Case-Control Study. J. Neurooncol. 2016, 130, 505-515. doi.org/10.1007/s11060-016-2244-4.
  11. Huang, X. Iron Overload and Its Association with Cancer Risk in Humans: Evidence for Iron as a Carcinogenic Metal. Mutat. Res. 2003, 533:153-171. doi: 10.1016/j.mrfmmm.2003.08.023.
  12. Rajaraman, P.; Schwartz, B.S.; Rothman, N.; Yeager, M.; Fine, H.A.; Shapiro, W.R.; Selker, R.G.; Black, P.M.; Inskip, P.D. Delta-Aminolevulinic Acid Dehydratase Polymorphism and Risk of Brain Tumors in Adults. Environ. Health Perspect. 2005, 113:1209-1211. doi: 10.1289/ehp.7986.
  13. Meng, Y.; Tang, C.; Yu, J.; Meng, S.; Zhang, W. Exposure to Lead Increases the Risk of Meningioma and Brain Cancer: A Meta-Analysis. J. Trace Elem. Med. Biol. 2020, 60, 126474. doi: 10.1016/j.jtemb.2020.126474.
  14. Pekic, S.; Stojanovic, M.; Popovic, V. Pituitary Tumors and the Risk of Other Malignancies: Is the Relationship Coincidental or Causal? Endocr. Oncol. 2022, 2, R1–R13. doi: 10.1530/EO-21-0033.
  15. Pepe, S.; Korbonits, M.; Iacovazzo, D. Germline and Mosaic Mutations Causing Pituitary Tumours: Genetic and Molecular Aspects. J. Endocrinol. 2019, 240 R21–R45. doi.org/10.1530/JOE-18-0446.
  16. Waalkes MP, Anver M & Diwan BA. Carcinogenic Effects of Cadmium in the Noble (NBL/Cr) Rat: Induction of Pituitary, Testicular, and Injection Site Tumors and Intraepithelial Proliferative Lesions of the Dorsolateral Prostate. Toxicol. Sci. 1999, 52, 154–161. doi.org/10.1093/toxsci/52.2.154.
  17. Kulas, J.; Tucovic, D.; Zeljkovic, M.; Popovic, D.; Popov Aleksandrov, A.; Kataranovski, M.; Mirkov, I. Aryl Hydrocarbon Receptor is Involved in the Proinflammatory Cytokine Response to Cadmium. Biomed. Environ. Sci. 2021, 34 192–202. doi.org/10.3967/bes2021.025.
  18. Caffo, M.; Barresi, V.; Caruso, G.; Cutugno, M.; La Fata, G.; Venza, M.; Alafaci, C.; Tomasello, F. Innovative Therapeutic Strategies in the Treatment of Brain Metastases. Int. J. Mol. Sci. 2013, 14, 2135-2174; doi:10.3390/ijms14012135.
  19. Kruger, A.; Sanchez-Sweatman, O.H.; Martin, D.C.; Fata, J.E.; Ho, A.T.; Orr, F.W.; Ruther, U.; Khokha, R. Host TIMP-1 Overexpression Confers Resistance to Experimental Brain Metastasis of a Fibrosarcoma Cell Line. Oncogene 1998, 16, 2419–2423. doi:10.1038/sj.onc.1201774.
  20. Haga, A.; Nagase, H.; Kito, H.; Sato, T. Enhanced Invasiveness of Tumour Cells After Host Exposure to Heavy Metals. Eur. J. Cancer 1996, 32A, 2342–2347. doi:10.1016/S0959-8049(96)00349-8.
  21. Wei, T.; Jia, J.; Wada, Y.; Kapron, C.M.; Liu, J. Dose Dependent Effects of Cadmium on Tumor Angiogenesis. Oncotarget 2017, 8, 44944–44959. doi:10.18632/oncotarget.16572.
  22. Mates, J.M.; Segura, J.A.; Alonso, F.J.; Marquez, J. 2010. Roles of Dioxins and Heavy Metals in Cancer and Neurological Diseases Using ROS-Mediated Mechanisms. Free Radic. Biol. Med. 2010, 49, 1328–1341. doi:10.1016/j. freeradbiomed.2010.07.028.
  23. Green, S.E.; Luczak, M.W.; Morse, J.L.; DeLoughery, Z.; Zhitkovich, A. 2013. Uptake, P53 Pathway Activation, and Cytotoxic Responses for Co(II) and Ni(II) in Human Lung Cells: Implications for Carcinogenicity. Toxicol. Sci. 2013, 136, 467–477. doi:10.1093/toxsci/kft214.
  24. Gaman, L.; Radoia, M.P.; Deliac, C.E.; Luzardo, O.P.; Zumbado, M.; Rodríguez-Hernándeze, A.; Stoiana, I.; Gilca, M.; Boada, L.D.; Henríquez-Hernández, L.A. Concentration of Heavy Metals and Rare Earth Elements in Patients with Brain Tumours: Analysis in Tumour Tissue, Non-Tumour Tissue, and Blood. Int. J. Environ. Health Res. 2021, 31, 741-754. doi.org/10.1080/09603123.2019.1685079.

An illustrative image should make the review easier to read.

In accordance with the reviewer's suggestion, we have added figure 1 in the revised text, section “Heavy Metals and Oxidative Stress”.

Figure 1: Schematic representation of heavy metals in relation to production of ROS.

Reviewer 2 Report

Review of the manuscript entitled: Impact of Heavy Metals on Gliomas Tumorigenesis.

The manuscript is interesting, the abstract and introduction are prepared correctly. The rest of the manuscript is very simple. In its current form, I do not recommend it for publication because it does not contribute much to knowledge.

Strongly recommends the authors to extend the work to include metal nanoparticles. Currently, most exposure to heavy metals comes from nanoparticles.

Author Response

We thank Reviewer for suggestions.

Strongly recommends the authors to extend the work to include metal nanoparticles. Currently, most exposure to heavy metals comes from nanoparticles.

In accordance with your suggestion, we, in the revised text, have added a new section “Metal Nanoparticles”.

Nanoparticles (NPs) are particles with at least one dimension < 1 μm, and potentially as small as atomic- and molecular-length scales (∼0.2 nm) [52]. Although NPs applications have great potentials, there are concerns about their adverse effects on human health and environment. The same properties (size, shape, and chemical composition) of the particle, can have an impact on the ecosystem [53]. Exposure to air pollution has numerous health effects, and due to the growing trend of pollutants. Environmental NPs are byproducts of combustion processes (combustion of fossil fuels, petroleum, wood burning), automobile exhaust, the activity of volcanoes or industrial derivatives [54]. Industrial NPs are frequently composed of heavy metals and their metal oxides [55]. Metal NPs can enter the body mainly through the respiratory, dermal, gastrointestinal, circulatory, immunological, and neurological tracts. The action of NPs is initially due to direct contact with cells (primary damage), causing toxicity, oxidative stress, DNA damage, inflammation. The NPs themselves, due to their peculiar properties, enter the blood circulation and, consequently, can also accumulate in the various organs, causing inflammation while altering, at the same time, their physiological functions (secondary damage). Environmental pollutants may reach the CNS through the transmission of NPs directly from the sinuses to the brain tissue through the olfactory nerves, from the passage of NPs through the alveoli, and the blood-brain barrier [56]. Toxicity from metal-NPs depend on its physical and chemical properties: molecular shape, size, oxidation status, surface area, bonded surface species, surface coating, solubility, and degree of aggregation cause to the generation of ROS and toxicity [57]. The induction of oxidative stress, as a consequence of the generation of ROS, activates a series of inflammatory mediators, such as nuclear factor-κB (NF-κb), mitogen-activated protein kinase (MAPK), and phosphoinositide 3-kinase (PI3K) pathway [59]. It has been reported that nano-CuO shows notable cytotoxic and DNA damaging capacity, leading to 8-hydroxy-20-deoxyguanosine (8-OHdG) formation [58]. Metal oxide NPs including ZnO, Fe3O4, Al2O3, and CrO3 of particle sizes ranging from 30 to 45 nm have been reported to induce apoptosis [60]. Cadmium telluride quantum dots were found to significantly increase p53 levels and upregulate the p53-downstream effectors Bax, Puma, and Noxa [61]. Metal oxide NPs such as cadmium, and iron have also been reported to show toxic effects via the induction of inflammatory-related cytokine release induced by NF-κB.

  1. Caruso, G.; Caffo, M.; Alafaci, C.; Raudino, G.; Cafarella, D.; Lucerna, S.; Salpietro, F.M.; Tomasello, F. Could Nanoparticle Systems Have a Role in the Treatment of Cerebral Gliomas? Nanomedicine 2011, 7, 744-752. doi: 10.1016/j.nano.2011.02.008.
  2. Caruso, G.; Marino, D.; Caffo, M. Nanoparticles and CNS Delivery of Therapeutic Agents in the Treatment of Primary Brain Tumors. J. Analyt. Oncol. 2014, 3, 105-112.
  3. Bystrzejewska-Piotrowska, G.; Golimowski, J.; Urban, P.L. Nanoparticles: Their Potential Toxicity, Waste and Environmental Management. Waste Manag. 2009, 29, 2587–2595. doi: 10.1016/j.wasman.2009.04.001.
  4. Zoroddu, M.A.; Medici, S.; Ledda, A.; Nurchi, V.M.; Lachowicz, J.I.; Peana, M. Toxicity of Nanoparticles. Curr. Med. Chem. 2014, 21, 3837–3853. doi: 10.2174/0929867321666140601162314.
  5. De Prado Bert, P.; Mercader, E.M.H.; Pujol, J.; Sunyer, J.; Mortamais, M. The Effects of Air Pollution on the Brain: A Review of Studies Interfacing Environmental Epidemiology and Neuroimaging. Curr. Environ. Health Rep. 2018, 5, 351–364. doi.org/10.1007/s40572-018-0209-9.
  6. Bantz, C.; Koshkina, O.; Lang, T.; Galla, H.J.; Kirkpatrick, C.J.; Stauber, R.H.; Maskos, M. The Surface Properties of NP Determine the Agglomeration State and the Size of the Particles Under Physiological Conditions. Beilstein J. Nanotechnol. 2014, 5, 1774-1786. doi: 10.3762/bjnano.5.188.
  7. Zhu, X.; Hondroulis, E.; Liu, W.; Li, C. Biosensing Approaches for Rapid Genotoxicity and Cytotoxicity Assays Upon Nanomaterial Exposure. Small 2013, 9, 1821-1830. doi: 10.1002/smll.201201593.
  8. Azad, M.B.; Chen, Y.; Gibson, S.B. Regulation of Autophagy by Reactive Oxygen Species (ROS): Implications for Cancer Progression and Treatment. Antioxid. Redox Signal. 2009, 11, 777-790. doi: 10.1089/ars.2008.2270.
  9. Poli, G.; Leonarduzzi, G.; Biasi, F.; Chiarpotto, E. Oxidative Stress and Cell Signalling. Curr. Med. Chem. 2012, 11, 1163-1182. doi.org/10.2174/0929867043365323.
  10. Choi, A.O.; Brown, S.E.; Szyf, M.; Maysinger, D. Quantum Dot-Induced Epigenetic and Genotoxic Changes in Human Breast Cancer Cells. J. Mol. Med. 2008, 86, 291-302. doi: 10.1007/s00109-007-0274-2.

Round 2

Reviewer 2 Report

The manuscript can be accepted.